# Estimation of extreme precipitations in Estonia and Italy using dual-pol weather radar QPEs

Roberto Cremonini [*1,3], Tanel Voormansik[2,4], Piia Post[2], and Dmitri Moisseev[1,5]

[1]Department of Physics, University of Helsinki, Finland
[2]Institute of Physics, University of Tartu, Estonia
[3]Regional Agency for Environmental Protection of Piemonte, Department for Natural and Environmental Risks, Torino, Italy
[4]Estonian Environment Agency, Tallinn, Estonia
[5]Finnish Meteorological Institute, Helsinki, Finland

**Correspondence:** Roberto Cremonini (rcremoni@ad.helsinki.fi)

**Abstract.** Evaluating extreme rainfalls for a certain location is commonly considered when designing stormwater management systems. Rain gauge data are widely used to estimate rainfall intensities for a given return period. However, the poor spatial and temporal resolution of operational gauges is the main limiting factor. Several studies have used rainfall estimates based on weather radar horizontal reflectivity ($Z_h$), but they come with a great caveat: while proven reliable on low or moderate rainfall rates, they are subject to major errors in extreme rainfall and convective cases. It is widely known that C-band weather radar can underestimate precipitation intensity due to signal attenuation or overestimate it due to hail and clutter contamination. Since the late 1990s dual-polarization weather radar started to become operational in the National Surveillance radar network in Europe, providing innovative QPE estimation based on polarimetric variables. This study circumvents $Z_h$ shortcomings by using specific differential phase ($K_{dp}$) data from operational dual-polarization C-band weather radars. The rain intensity estimates based on specific differential phase data are immune to attenuation and less affected by hail contamination.

In this study, for the first time, quantitative precipitation estimations (QPEs) based on polarimetric observations by operational C-band weather radars and without any raingauges adjustments are analyzed. The purpose is to estimate return periods for 1-hour rainfall total computed from polarimetric weather radar data using non-adjusted QPEs based on $R(Z_h, K_{dp})$ data and to compare the results with those ones derived using $R(Z_h)$ and rain-gauge data. Only the warm period during the year is considered here, as most of the extreme precipitation events for such duration take place for both places at this time. Limiting the dataset to warm periods also allows us to use the radar-based rainfall quantitative precipitation estimates, which are more reliable than the snowfall ones. Data from operational dual polarimetric C-band weather radar sites are used both from Italy and Estonia. Given climatological homogeneous regions, this study demonstrates that polarimetric weather radar observations can provide reliable QPEs compared to single-polarization estimates with respect to rain gauges and, that can provide a reliable estimation of return periods of 1-hour rainfall total, even for relatively short time series.

*Corresponding author: Roberto Cremonini, Department of Physics, University of Helsinki, Finland, e-mail: rcremoni@ad.helsinki.fi

# 1 Introduction

The increase in impervious surfaces due to urbanization leads to an increase in flooding frequency due to poor infiltration and faster concentration time. The hydrological changes, driven by heavy urbanization, and resulting impacts on extreme rainfall, are also being established: a significant amount of research over the last twenty years has shown a strong relationship between urban areas and local microclimate.

The IPCC Sixth Assessment Report (IPCC, 2021) increased the interest on short-duration rainfall extremes estimations as several Earth regions are likely to be affected by an increase in heavy precipitation events in the near future due to global warming. In Europe van den Besselaar et al. (2012) demonstrated that higher latitudes are yet experiencing an increment in intensity and frequency of extreme events, and correspondingly in heavy precipitations. For all these reasons, studies on extreme annual rainfall maximum depths for short durations are extremely relevant for hydrological studies, water management, and urban areas development (Marra et al., 2017).

However, the reliability of traditional rainfall depths estimations is often limited by the low spatial density of rain gauge networks, particularly for short durations (Overeem et al., 2010). Nevertheless, single-polarization weather radars can provide quantitative precipitation estimates (QPEs), based on empirical relationships between the equivalent reflectivity factor at horizontal polarization ($Z_h$) and the rain rate with proper spatial and temporal resolution. Several studies investigated statistics of extreme areal rainfall depths obtained from single-polarization weather radar (Frederick et al., 1977; Allen and De Gaetano, 2005; Overeem et al., 2008, 2009a, b, 2010; Marra and Morin, 2015; Panziera et al., 2018; Marra et al., 2022). Keupp et al. (2017) and Fabry et al. (2017) offer a complete review of monthly or annual rainfall climatology based on weather radar observations respectively in Europe and the contiguous United States (CONUS) area.

However, due to signal attenuation at C-band (Delrieu et al., 2000) and due to hail contamination (Ryzhkov et al., 2013), the horizontal radar reflectivity ($Z_h$) is subjected to significant errors, especially during intense rainfalls and convective precipitations. As stated by Fairman et al. (2015), relevant QPEs underestimations typically occur in mountainous areas and far away from the weather radar; beam-blocking and overshooting also cause large differences between radar-based QPEs and reference gauges. To overcome these limitations, several adjustment techniques have been developed, correcting QPEs, derived from single-polarization weather radar, with raingauges measurements (Einfalt and Michaelides, 2008; Goudenhoofdt and Delobbe, 2009). Overeem et al. (2009b) derived short-duration extreme rainfall depths from gauges-adjusted weather radar QPEs. Barndes et al. (2001), Ryzhkov et al. (2005) and Vulpiani et al. (2012) demonstrated that polarimetric rainfall estimation algorithms based on specific differential phase ($K_{dp}$) outperform the conventional QPEs based on horizontal radar reflectivity, being immune from partial beam-blocking, attenuation, hail contamination, and weather radar miscalibration. Several studies focused on the evaluation of $R(K_{dp})$ relationships performances with respect to traditional $R(Z_h)$ for precipitation events (Paulitsch et al., 2009; Moisseev et al., 2010; Cremonini and Bechini, 2010). Voormansik et al. (2021a) deeply analyzed five years QPEs derived from operational C-band polarimetric weather radar in Estonia and Italy, demonstrating that $R(Z_h, K_{dp})$ blended algorithms provide good quality QPEs.

For the first time, this study investigates the statistical properties of annual rainfall maximum for 1-hour rainfall total, analyzing

QPEs derived from $R(Z_h, K_{dp})$ observations by operational dual-polarization C-band weather radars in two different climate regions. The results from short-period weather radar observations are compared with statistics obtained from gauges measurements and QPEs based on traditional horizontal radar reflectivity. Section 2 provides a description of study areas, polarimetric weather radar systems, and algorithms used to derive QPEs. In Section 3 extreme value statistic is applied to fit the theoretical distributions, providing rainfall depth as a function of duration for given return periods. Finally, discussion and conclusions follow.

## 2 Materials and methods

This study focuses on QPEs based on polarimetric C-band weather radar, operating in Northern Italy and Estonia. The studied period is limited to the warm period of the year as most of the extreme precipitation events at short temporal scales take place at this time. Limiting the dataset to a warm period also helps to exclude that weather radar observations come from the snow or ice crystals, a requirement for reliable rainfall intensity estimations based on $R(Z_h, K_{dp})$.

### 2.1 The study areas

This study focuses on areas in Piemonte, Italy, and Estonia, covered by operational dual-polarization Doppler C-band weather radars operated by the local weather services.

Piemonte is located in northwestern Italy, in the upper areas of the Po valley; the central part of the region is relatively flat (300-200 m a.s.l.) with the Torino hill that reaches 770 meters a.s.l.. The Alps surround plains with altitudes ranging from 1,000 m to more than 4,500 m a.s.l.. The two areas considered in this study are centred on Torino hill and they extend for about 30-50 km far from the weather radar, corresponding to about 7,300 $km^2$ altogether (Figure 1, the left map). To ensure QPEs data quality, the choice to restrict the study areas close to the radar site is driven by these main reasons:

1. to reduce weather radar beam-broadening and beam-propagation effects;

2. to avoid the Alps complex orography in western and northern directions;

3. to limit the weather radar beam height above ground;

4. to avoid or to limit spatial non-stationarity of the GEV parameters and their dependence from geomorphology (altitude, terrain slope and exposition).

The Piemonte rainfall regime is sub-continental with a dry season during winter, the main maximum precipitation occurs during fall and a secondary maximum during spring-summer (Devoli et al., 2018); convective precipitations are very frequent from late spring to early fall. Pavan et al. (2018) reconstructed rainfall climatology over Po valley from gauges observations from 1961 to 2015, showing that, although the relatively small extent of the whole study areas, there are different precipitation regimes between the area A1, located close to the Alps, (wetter) and the area A2, the flats south of Torino hill, (dryer). It is worth mentioning that the average annual rainfall within each single study area is uniform.

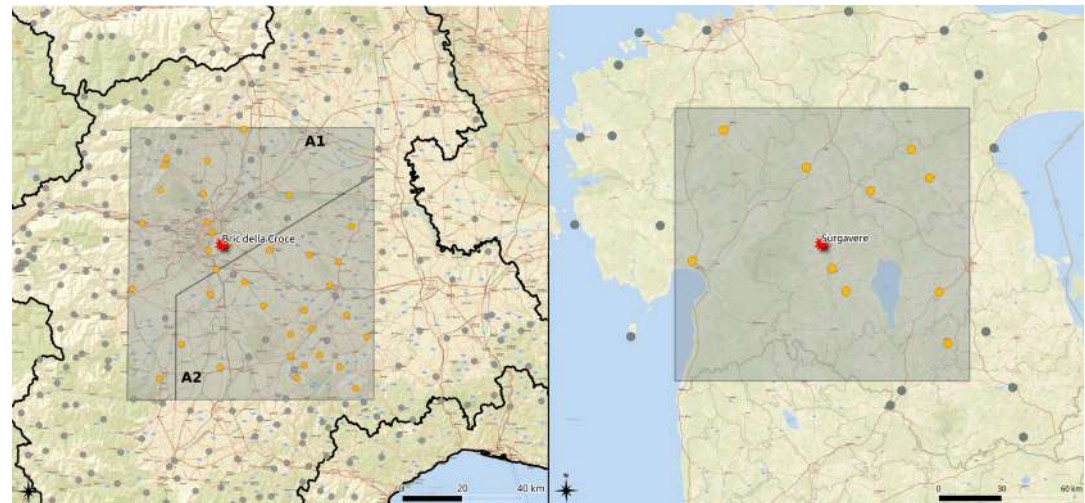

**Figure 1.** The study areas. On the left the two Italian study areas (EPSG:32632) and on the right the Estonian area (EPSG: 3301). The dots symbols show the tipping-bucket raingauges of the local hydrological networks, orange dots symbols are the tipping-bucket raingauges used in the study, the red stars are the weather radar location, respectively the Bric della Croce radar site and Sürgavere radar location; basemap: ESRI, https://basemaps.arcgis.com/arcgis/rest/services/World_Basemap_v2/VectorTileServer accessed on 5 November 2022

85  The Bric della Croce weather radar, operated by the regional agency for environment protection (Arpa Piemonte) is located on the top of Torino hill. The operational radar completes fully polarimetric volume scans, made of eleven elevations up to 170 km range with 340 m range bin resolution. Quantitative precipitation estimates (QPEs), based on horizontal reflectivity, are extensively described by Cremonini and Tiranti (2018), meanwhile, $K_{dp}$ based precipitation estimations are derived according to Wang and Chandrasekar (2009). The closest observations to the weather radar (up to eight kilometres) have been left out

90 due to heavy ground clutter contamination and unreliable estimations of $K_{dp}$. Being focused on convective precipitation, this study limits the analysis to the warm season ranging in Italy from April to October. Bric della Croce data range from 2014 to 2020 with five minutes interval time resolution. The data inspection for quality purposes has shown that the annual maxima for the years 2015 and 2016 are unreliable, due to frequent weather radar failures during the warm season: for this reason, these years have been excluded from the following analysis.

95 Arpa Piemonte also operates an automated ground weather network made by more than 350 raingauges with 0.2 mm resolution and 300 mm/h maximum detectable rainfall intensity: one-minute rainfall observations are available since 1988. Annual hourly rainfall maxima are derived from gauges observations corrected for underestimations at high rainfall intensities according to Lanza et al. (2010) and Vuerich et al. (2009). Annual hourly rainfall maxima are manually quality controlled to identify possible mechanical failures and incomplete time series. In this work, one-minute resolution tipping-bucket raingauges located within

100 the two study areas and running for at least 15 years have been used. Area A1 north and west of the weather radar site contains 27(24) gauges, while area A2 contains 25(23) gauges; the annual hourly precipitation maxima concerns years from 1988 to 2020.

The study area in Estonia is centred on the continental part of the country and it extends for about 70 km around the radar corresponding to 16,911 $km^2$. Estonia is a flat country with a mean elevation of about 50 m a.s.l. and the highest point being 318 m a.s.l in the more hilly southeast (Figure 1, right map). Estonia has a temperate climate with the heaviest rainfall in late summer. Convective precipitation is common in the area from May to September (Voormansik et al., 2021b). There are distinct differences in precipitation climate between continental Estonia and the islands in the western part as the latter is much drier (Tammets and Jaagus, 2013). This variance is caused by different thermal regimes of sea and land surfaces. Sub-daily rainfall extremes in the Nordic-Baltic region from rain gauges were analysed by Olsson et al. (2022). Variations in 1-hour return levels in Estonia were found in some of the stations outside of our study area. Generally higher return levels were found in the eastern part of the country and lower return levels in the western part. The 1-hour return levels were shown to be nearly uniform in the central part of Estonia without significant variations of the shape parameter (see Section 2.1.1), being close to zero. In the study area, we can thus expect a uniform precipitation regime.

Sürgavere radar is situated in the northern part of Sakala upland on top of Sürgavere hill (128 m a.s.l.). Sürgavere radar has been operational since 2008 and a continuous archive is available since 2010. Until May 2020, the radar performed a volume scan with eight elevations up to 250 km range with 300 m range bin resolution every 15 minutes. In May 2020 the scan strategy received a major update. Since then the radar scans seven elevations with a 250 km range every five minutes and the lowest elevation with a 250 km range every 2.5 minutes. After careful inspection of reflectivity and polarimetric data quality, five years of radar data (2012-2013 and 2018-2020) were included in the study. Data from 2014, 2015, and 2017 were not included because of insufficient polarimetric data quality to obtain reliable QPEs. 2014 and 2015 were excluded because of a broken waveguide limiter which caused a gradually decreasing polarimetric data quality. Data from 2017 was left out because a broken stable local oscillator (STALO) reduced the data quality to levels not usable for QPE purposes. The year 2016 was omitted because of the low availability of radar data due to frequent and long-lasting radar failures (availability of 30% for August and 85% for the whole summer period of that year) that would result in unreliable annual maxima. Mean radar data availability for the investigated five-year period was 98%. Only 15-minute interval data are used in this study to maintain homogeneity.

$K_{dp}$ precipitation estimates of Estonia are derived using PyART function *phase_proc_lp* (Giangrande et al., 2013). Compared to the work by Voormansik et al. (2021a) done in the same study area some parameters of this function have been changed. The necessity of updating the parameters became inevitable because using the parameters of the earlier work led to unrealistically high 1-hour rainfall maxima and over-smoothed precipitation fields. The parameters of the function that were changed were *window_len*, *high_z*, and *coeff*. The first of these, *window_len*, allows changing the length of the Sobel window applied to $\Phi_{dp}$ field before calculating $K_{dp}$. When using the default window length of 35 bins (equal to around 10.5 km in our case of 300 m bins), the function produces less accurate results in $K_{dp}$ fields with steep gradients and large $K_{dp}$ magnitudes as it oversmooths the $\Phi_{dp}$ field (Reimel and Kumjian, 2021). We tested with various window lengths and found the length of 8 bins (equal to 2.4 km in our case) to be the optimal compromise between spatial resolution and smoothness. After the window length change, we obtained realisticlooking precipitation fields but the overestimation compared to gauge values increased. This is because $\Phi_{dp}$ gradients became steeper due to the smaller window length. To mitigate this issue we first decreased the *high_z* (the high limit for reflectivity to remove hail contamination) value from 60 dBZ used in Voormansik et al. (2021a) to 50 dBZ which is the

lowest recommended value by Giangrande et al. (2013). Because overestimation was still evident we also reduced the $Z_h$-$K_{dp}$ self-consistency coefficient. As stated by Kumjian et al. (2019) the $Z_h$-$K_{dp}$ consistency relationships probably do not exist in hail and it is therefore recommended to reduce the weight of the self-consistency constraint in the case of hail (Reimel and Kumjian, 2021). We tested with various values and found a coefficient value of 0.9 to produce optimal results.

The following equations have been used to derive rain rate from weather radar variables:

$$R(Z_h) = 300Z^{1.5} \quad \text{from Joss and Waldvogel (1970)} \tag{1}$$

and

$$R(K_{dp}) = 21.0K_{dp}^{0.720} \quad \text{from Voormansik et al. (2021a)} \tag{2}$$

Horizontal reflectivity data are re-calibrated using a method that makes use of the knowledge that $Z_h$, $Z_{dr}$ (differential reflectivity), and $K_{dp}$ are self-consistent with one another and one can be computed from two of the others. The calibration was carried out using the theory set down in Gorgucci et al. (1992) and Gourley et al. (2009) where the process is described in detail. As a result, $Z_h$ bias of 2.0 to 5.0 dB depending on the data period is obtained and added to the corresponding original reflectivity data. Data up to 10 km from the radar were excluded because of the ground clutter and unreliable $K_{dp}$ estimation. Weighted rain gauges operated by the Estonian Environment Agency (EstEA) located in the study area are used as ground truth to compare with radar estimates. The rain gauges provide data with a resolution of 0.1 mm and maximum detectable rainfall intensity of 2000 mm/h. Rainfall observations from 2003-2010 are available with a 1-hour resolution and starting from 2011 with 10-minute intervals. In this study, ten years of gauge data from 10 stations located in the study area from 2011 to 2020 are used. As demonstrated by Voormansik et al. (2021a), the combined product $R(Z_h, K_{dp})$ outperforms with respect to QPEs based on $R(Z_h)$ and $R(K_{dp})$. The weather radar-based QPE here used is defined as:

$$R(Z_h, K_{dp}) = \begin{cases} R(Z_h), & \text{if } Z_h \leq 25 \quad \text{dBZ} \\ R(K_{dp}), & \text{otherwise} \end{cases} \tag{3}$$

The evaluation of the horizontal reflectivity threshold has been derived by optimizing results on 1-hour accumulation rainfall in both locations, Italy and Estonia (Voormansik et al., 2021b).

### 2.1.1 Data quality, homogeneity, and goodness-of-fit

Both in Estonia and Italy, the annual 1-hour maxima derived from raingauges are manually quality controlled respectively by EstEA and Arpa Piemonte staff to identify possible technical issues or incomplete time series. L-moments are linear functions of sampling data and they are related to probability-weighted moments by the equation. With respect to conventional moments,

L-moments are more robust to outliers and enable more secure inferences to be made from small samples about an underlying probability distribution, suffering less from the effects of sampling variability. According to (Hosking and Wallis, 1997), the discordancy measure $D-i$ indicates, for site $i$, the discordancy between the site's L-moment ratios and the (unweighted) regional average L-moment ratios. Large values might be used as an indication of potential errors in the data at the site. The discordancy analysis has been performed on ground data for the three areas, identifying anomalous raingauges in Italy. They

correspond to gauges located in road bridges or nearby trees, where the local environment affects rainfall measurements. Those raingauges have been excluded from the following analysis.

Hosking and Wallis (1997) also recommend merging observations from the individual raingauges which come from homogeneous regions: homogeneity implies that a scaled data series has the same statistical distribution. The convective characteristics of the 1-hour annual maximum precipitation, here investigated, and its weak intrinsic spatial correlation also supports this

hypothesis for the study areas. Homogeneity tests have been also applied to evaluate the statistical coherence of the two study areas. Broadly used procedures for testing for regional homogeneity assessment are described and compared in Viglione et al. (2007). Hosking and Wallis (1997) proposed to test the homogeneity of pooled sites by a measure based on L-moment ratios, which compare the between-site variation in sample L-Cv (coefficient of a variation) values with the expected variation for a homogeneous pooling group. According to the L-moments used in the definition of the test statistics (H), they defined three hetero-

geneity measures: $H_1$ — when L-Cv is used, $H_2$ — if the L-Cs is used and $H_3$ — if the L-Ck is applied. If $H_i$ (i= 1, 2, 3) are less than one, then the region is *"acceptable homogeneous"*, if they are comprised between 1 and 2, then *"possibly homogeneous"*, else *"heterogeneous"*. By using the R-package *lmomRFA* (https://cran.r-project.org/web/packages/lmomRFA/index.html), the L-moment homogeneity test has been applied in all study areas, both in Italy and Estonia, obtaining $H_i$-values less than one, considered as *"acceptably homogeneous"*. This finding can be explained by the relatively small extension of the study areas

considered (about $60 \times 60 \, km^2$ for each study area) and by their homogeneity in terms of precipitation regimes and geomorphological characteristics.

Identifying the best probability distribution for describing the behaviour of the annual maxima data is one issue. Plots of L-Skewness and L-Kurtosis values both from raingauges and weather radar for the three study areas with a 1-hour duration

have been elaborated. In order to select the appropriate frequency distribution function, the L-moment ratio diagrams method has been used. The L-moment ratio diagram is a widely used tool for the graphic interpretation and comparison of the sample L-moment ratios, L-Cs (skewness), and L-Ck (kurtosis) of various probability distributions (Hosking and Wallis, 1997). Figure 2 shows the L-moment ratio diagram for raingauges (left) and for weather radar-based QPEs (right). The right panels of L-moment ratios have been derived from sampling weather radar-based QPEs 500 times by random uniform sample. The

closeness of the regional mean and the at-site data for the three study areas and for raingauges and weather radar-based data to the GEV distribution is evident. Accordingly, a goodness-of-fit test statistic (Hosking and Wallis, 1997) was used in identifying the best three-parameter theoretical distribution. The goodness-of-fit test is based on a comparison between the L-Ck sample and the L-Ck population for different distributions. An acceptable distribution function should achieve a value of

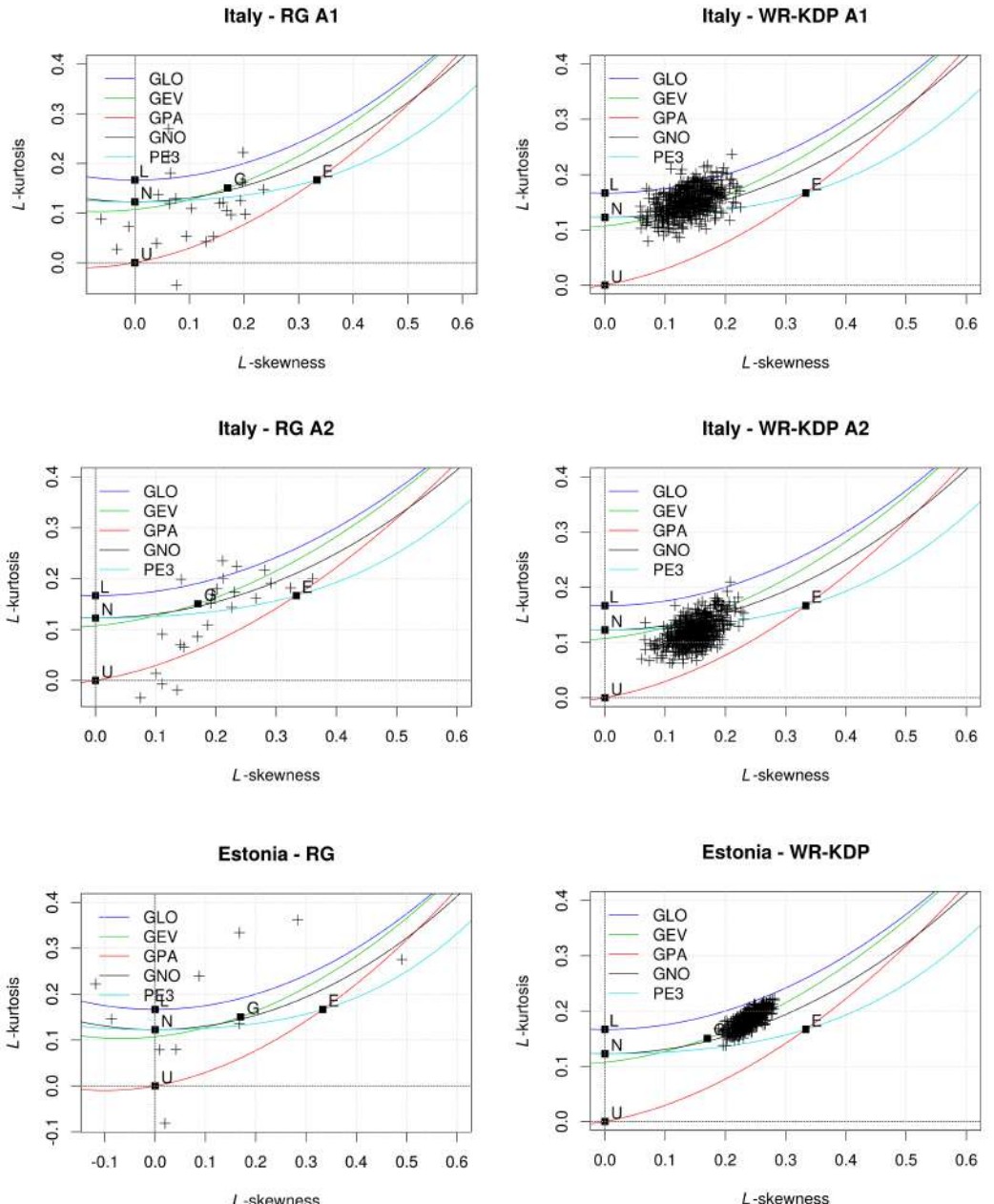

**Figure 2.** L-moment ratio diagrams for raingauges (left) and for QPE weather radar-based (right).

$Z_{DIST} \leq 1.64$. For all the datasets of 1-hour annual precipitation maxima derived from raingauges or from weather radar, the
Gumbel distribution is acceptable.

### 2.1.2 Extreme-value distributions

The statistics of extreme values describe the behaviour of the largest of $m$ values: large and consequently rare values are considered extreme. The fundamental result from the theory of extreme-value statistics asserts that, regardless of the (single, fixed) distribution from which the observations have come, the largest of $m$ independent observations from a fixed distribution will match a known distribution more closely as $m$ increases (Extremal Types Theorem, Coles (2001)). Extensive literature, dating back to the 1940s, deals with extreme values theory in its formalization and its hydrological applications: the introduction theory and a historical review on this topic can be found in Papalexiou and Koutsoyiannis (2013), Wilks (2011), De Haan and Ferreira (2006).

Given $R_{1h}$ the random variable of annual maximum rainfall accumulation for the hourly duration, the cumulative distribution function is given by $F(z)$ (Jenkinson, 1955):

$$F(R_{1h} \leq z) = \exp(-[1 + \xi \frac{z - \mu}{\sigma}]^{-\frac{1}{\xi}}) \quad \text{with} \quad -\infty < \xi < +\infty, \, \sigma > 0, \, -\infty < \mu < +\infty \quad (4)$$

defined as $z : 1 + \xi(x - \mu)/\sigma > 0$, where $\mu$, $\sigma$ and $\xi$ are the location, scale and shape parameters, respectively.

According to Katz et al. (2002), the GEV distribution, which combines three different statistical families (Gumbel, Fréchet, and Weibull), can fit the extreme data set with high accuracy.

The GEV distribution unites the Gumbel, Fréchet, and Weibull distributions into a single family to allow a continuous range of possible shapes (Frűh et al., 2010; Coles, 2001). These three distributions are known as type I, II, and III extreme value distributions. The GEV distribution is parameterized with a location parameter ($\mu$), scale parameter ($\sigma > 0$), and shape parameter ($\xi$). The GEV distribution is equivalent to Type I, II, and III, respectively, when the shape parameter is equal to zero, greater than zero, and lower than zero:

- – $\xi > 0$ Fréchet distribution (EV2);

- – $\xi = 0$ Gumbel distribution (EV1);

- – $\xi < 0$ Weibull distribution (EV3).

Based on the extreme value theorem, the GEV distribution is the limit distribution of properly normalized maxima of a sequence of independent and identically distributed random variables. Thus, the GEV distribution is used as an approximation to model the maxima of long (finite) sequences of random variables.

Several methods have been developed for the estimation of GEV distribution parameters, including the method of moments (MME), the method of L-moments (LME), the method of probability-weighted moments (PWME), and the method of maximum likelihood (MLE) (Katz et al., 2002; De Haan and Ferreira , 2006). Hereafter, the only MLE method has been used to estimate GEV distribution parameters from sample data.

By inverting Equation 4, an estimation of extreme quantiles can be obtained being the $p^{th}$-upper quantile of the z distribution

given by $F(z_p) = 1 - p$ where $z_p$ is the return level correlated to the return period T = 1/p. The $z_p$ versus 1/p plot is an effective tool to graphically observe the return levels and it is known as the return level plot.

The shape parameter controls the upper-tail behaviour, but it remains difficult to estimate on the basis of short time-series data (a few decades for example): it happens because there are usually few extremes exhibiting much variability. As stated by Lazoglou et al. (2018), the Weibull (negative shape parameter) is not appropriate for precipitation datasets.

The weather radar-based rainfall annual maxima statistics over the Netherlands calculated by Overeem et al. (2009a) have shown that regional differences in the location parameter exist for the most durations. Nevertheless, due to the small number of rainfall annual maxima, when depth-duration-frequency (DDF) curves are derived for small areas, the uncertainties in the DDF curves generally become larger compared to the uncertainties of the average DDF curve for the Netherlands. Recalling an increase in the standard errors of the quantile estimates, in particular at high return periods, Buishand (1991) underlies that large standard errors are mainly caused by the uncertainty in the shape parameter. On the other side, real-time operational applications, like issuing of early warnings, are based on relatively low quantiles (typically a 10-20 years return period). Within this range of return periods, the inaccuracy is expected to be considerably reduced (Marra et. al., 2019): for these reasons in this study, the Gumbel distribution ($\xi = 0$) have been assumed appropriate for 1-hour accumulation annual rainfall maxima.

As discussed by Overeem et al. (2009a), the spatial correlation of measurements affects extreme values statistics, carrying to underestimation. The correlation between two raingauges is typically low for convective precipitation, due to the small spatial scales involved in convection ($\approx 10 - 100\,km^2$) and the low density of the ground meteorological network (typically an order of one gauge every $100\,km^2$). In the case of weather radar observations, given the higher spatial resolution ($\approx 1\,km^2$), the correlation between close cell grids must be estimated and taken into account. Assuming GEV distribution parameters are constant in each of the areas considered in this study, their estimation from all data in the regions justifies the derivation of return periods longer than the rainfall record (Overeem et al., 2010). This statement assumes that both sample data are independent and the precipitation regime in the studied area is uniform. To avoid data spatial correlations, this study merely investigates 1-hour rainfall totals, disregarding longer durations.

Semi-variograms are widely used in geostatistic sciences for evaluating rainfall spatial structure. Semi-variograms summarize the spatial relations in the data, and they can be used to understand within what range data are spatially correlated (Naimi et al., 2011).

In this study, the statistical analysis has been conducted using the R (https://cran.r-project.org/) package ExtRemes 2.1 (Gilleland et al., 2016). The experimental isotropic semi-variogram can be derived by taking half the average of the squared difference between data pairs at equal distances and by assuming stationarity and isotropy of the rainfall field (Cressie, 1993):

$$\gamma(h) = \frac{1}{2n(|h|)} \sum_{k=1}^{n(|h|)} (z(x_k + h) - z(x_k))^2 \qquad (5)$$

where $x_k$ is the location of cell barycentre $k$ and $x_k + h$ is the location at distance $h$ from location $x_k$.

Figure 3 shows semi-variograms, obtained from $R(Z_h, K_{dp})$ annual hourly rainfall maxima, from April to September in Italy

for area A1 (left) and Estonia (right).

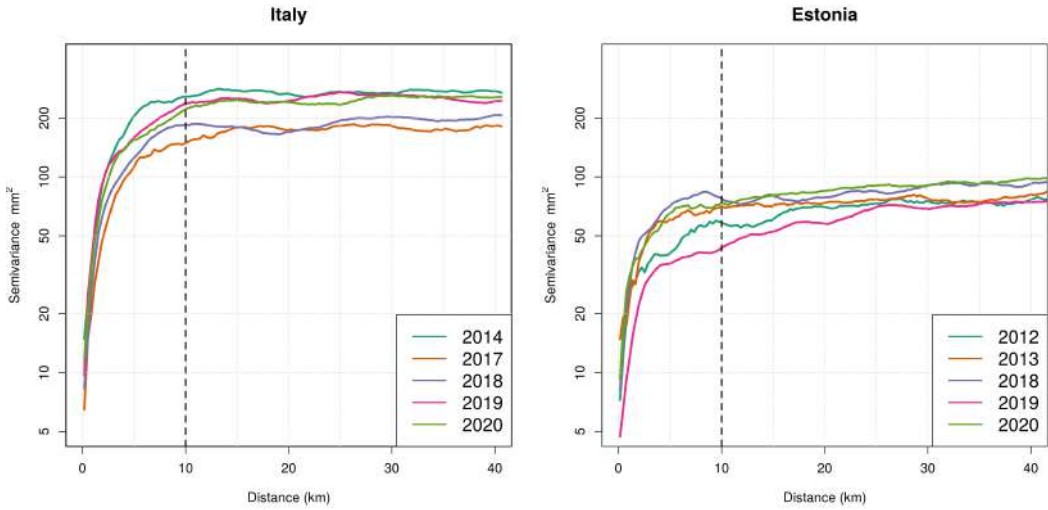

**Figure 3.** Empirical variograms for hourly rainfall annual maxima based on $R(Z_h, K_{dp})$ hourly rainfall estimations in Italy area A1 (left) and Estonia (right).


The empirical semi-variogram analysis for weather radar observations indicates that hourly rainfall maxima decorrelate at about 10 km both in Estonia and Italy (Figure 3). These results are consistent with past studies (Schroeer et al., 2018; Dzotsi et al., 2013): convective precipitation is prevalent during the warm season and, consequently, the spatial correlation quickly decreases with the distance between two rain gauges. Moreover, ten kilometres is the typical spatial scale of convective

precipitation systems ($meso - \gamma$). Different values of semi-variances in Estonia and Italy can be explained by the different climatic regimes, with generally weaker convective precipitations in the Baltic country. Hence, to avoid statistical oversampling and to ensure statistical independence of data samples, 1-hour rainfall total annual maxima estimated by weather radar are re-sampled according to the found spatial scale of convective precipitation. The hourly annual rainfall maxima estimated by weather radar observations are up-scaled from the original data resolution (340 meters for Italy and 300 meters for Estonia) to

10 km resolution, using a uniform random sampling algorithm.

## 3 Results

On the basis of the goodness-of-fit results, the observations relative to all three study areas have been fitted with Gumbel distributions. For weather radar QPEs, the fit has been performed considering the mean L-moments, derived from sampling random sampling 500 times the original data at 10 km resolution. Figure 4 shows the diagnostics from the Gumbel distribution fitted to

1-hour rainfall total annual maxima in Italy for area A1 (upper) and Estonia (lower), derived from $R(Z_h, K_{dp})$ estimates; from

left to right, the Figure shows the density plot of the data along with the model fitted density, the Q-Q plot of the data quantiles against the fitted model quantiles with 95% confidence bands, a Q-Q plot of quantiles from model-simulated data against the data. Quantile-quantile scatterplots compare empirical data and fitted CDFs in terms of quantiles: in an ideal perfect fitting, all points should lay on the 1:1 diagonal line (Wilks, 2011).

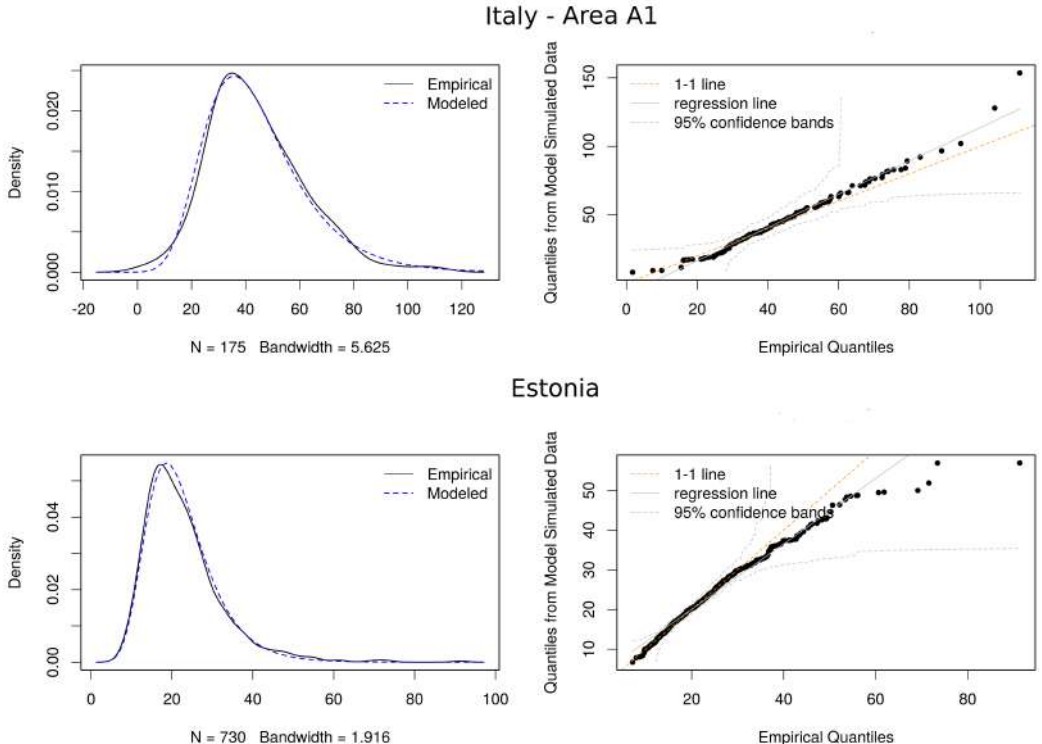

**Figure 4.** Diagnostic plots for 1-hour annual rainfall maxima fits derived by weather radar in Italy area A1 (upper) and Estonia (lower): from left to right, density plot of the data along with the model fitted density, Q-Q plot of the data quantiles against the fitted model quantiles with 95% confidence bands, a Q-Q plot of quantiles from model-simulated data against the data.

The Q-Q plots present some departures from linearity in correspondence with the tails, especially for Estonia data, which are due to the increasing level of uncertainty that characterizes model extrapolation at high levels. The empirical estimates in the return level plot reflect results in Q-Q plots laying very close to the model-based line, which results be almost linear, for low values. However, even if the return level estimates seem convincing, the increasing confidence bands for large return periods indicate the uncertainty that affects the model at high levels.

Table 1 summarizes the results of fitting data samples with Gumbel distribution by applying the Maximum Likelihood Estimation method (MLE) for each studied area; location and scale parameters with their standard errors ($\sigma/sqrt(n)$) are shown. Kolmogorov-Smirnov tests (Ks-test) and Q-Q plots inspections (Wilks, 2011) show that fits for $R(Z_h, K_{dp}$ a more

| Area | Source | $\mu$ (mm) | $\sigma$ | n-values |
|---|---|---|---|---|
| | WR-KDP | $35.7 \pm 1.0$ | $15.1 \pm 0.7$ | 175 |
| Italy - A1 | WR-ZH | $19.2 \pm 0.7$ | $9.5 \pm 0.6$ | 175 |
| | RG | $30.1 \pm 0.5$ | $11.4 \pm 0.4$ | 550 |
| | WR-KDP | $33.40 \pm 0.9$ | $11.3 \pm 0.7$ | 171 |
| Italy - A2 | WR-ZH | $18.0 \pm 0.6$ | $7.7 \pm 0.5$ | 171 |
| | RG | $25.3 \pm 0.4$ | $9.8 \pm 0.3$ | 484 |
| | WR-KDP | $17.4 \pm 0.2$ | $5.9 \pm 0.1$ | 800 |
| Estonia | WR-ZH | $14.6 \pm 0.3$ | $6.8 \pm 0.2$ | 800 |
| | RG | $15.1 \pm 0.6$ | $5.9 \pm 0.5$ | 93 |

**Table 1.** Estimated Gumbel parameters, location and scale $(\mu, \sigma)$ for weather radar and gauges annual hourly maxima rainfall intensities for Italy area A1 and area A2 and Estonia for weather radar derived from $R(Z_h, K_{dp})$ (WR-KDP), derived from $R(Z_h)$ (WR-ZH) and raingauges (RG) time-series observations

reliable than $R(Z_h)$ ones in all the three study areas. This behaviour is particularly evident for the Estonia area, where the fit distribution for $R(Z_h)$ leads to a large value of the scale parameter and an anomalous low p-value obtained by KS-test.

It is well-known that record length affects the estimate of the GEV shape parameter and long historical time series are needed for reliable estimates. Papalexiou and Koutsoyiannis (2013), Ragulina and Reitan (2017), Lazoglou et al. (2018), Lutz et al. (2020), Deidda et al. (2021) demonstrated that the shape parameter tends to have positive values, between 0 and 0.23 with a probability of 99%, as sample size increases. However, in this study, given the shortness of the weather radar time series, for safety the scale parameter has been set to zero, according to Papalexiou and Koutsoyiannis (2013).

## 4   Discussion


Several studies developed adjustment techniques to correct QPEs based on weather radar observations with raingauge measurements (Einfalt and Michaelides, 2008; Goudenhoofdt and Delobbe, 2009). For the first time, this study investigates extreme precipitation estimation using dual-polarization weather radar rainfall estimations without any adjustment with raingauges. It is worth recalling that the study has been limited to relatively flat and geomorphologically homogeneous areas with high-quality

dual-polarization weather radar observations close to the ground and high-quality 1-hour rainfall total annual maxima from raingauges. Weather radar data quality and reliability have been carefully checked in Voormansik et al. (2021a).

The two studied regions, Estonia and Italy, are characterized by different precipitation regimes, colder the first one and warmer the latter. The different climate regimes of the studied areas consequently reflect on fitted Gumbel distributions, determining lower return periods in Italy, given a 1-hour rainfall total. Estonia is characterized by few rain gauges and by a limited historical

series, but also by a larger homogeneous flat region covered by the operational polarimetric weather radar. In this area, it can be appreciated the benefit of estimating Gumbel distribution using weather radar observations after ensuring spatial independence

and assumed homogeneity: the sample size derived from five years of observations is about nine times the sample size obtained by raingauges. These different sample sizes determine larger standard deviations in Gumbel distribution parameters estimation by raingauges with respect to weather radar-based estimations.

In Italy, a dense automatic gauges network is operating since 1988, providing about 25 gauges per area, determining a larger sample size. But, the Alps and the spatial variability of the climate regime, influenced by a complex orography, limit the availability of high-quality weather radar observations to about 160-180 values. Despite the limited availability of weather radar observations (only five years for both Italian and Estonian weather radars), the comparison of Gumbel distribution fits in these two different regions has shown encouraging results.

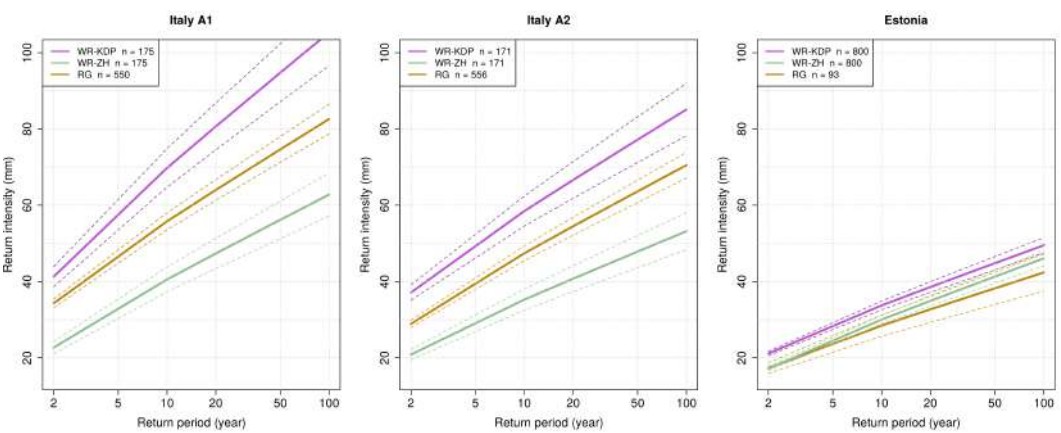

**Figure 5.** Return levels for 1-hour rainfall accumulation in Estonia (left) and Italy area A1 (centre) and area A2 (right) derived from the Gumbel distributions. The dash lines show confidence intervals for $\alpha = 0.05$.

Figure 5 shows return levels for 1hour rainfall at a given return time, estimated from Gumbel distributions with location and shape parameters from Table 1 for the study areas and estimated from $R(Z_h)$, $R(Z_h, K_{dp})$ and from raingauges.

In Italy, the different return periods between the two areas are in agreement with findings in Mezzoglio et al. (2022) and with climate classification of the two areas reported by Pavan et al. (2018), with area A1 more favourable to intense precipitation than area A2. This precipitation regime, confirmed also by climatological lightning density (not shown), can be justified by

local geomorphology. In fact, during the warm season, cold air overcomes the Alps flowing towards the Po valley from west-northwest: the Monferrato hills east of Torino enhance lowlevel convergences and strong uplifts, causing deep convection in area A1, while study area A2 experiences downwind conditions. QPEs based on $Z_h - K_{dp}$ algorithm generally provided slightly shorter return periods with respect to gauges estimations: this behaviour can be explained by the high spatial resolution of weather radar observations able to catch small-scale rain showers. However, it could be due to a slight overestimation of annual

rainfall maxima by weather radars, as highlighted by Voormansik et al. (2021a), and hence needing further investigations. The distribution fits based on $R(Z_h)$ show a longer return period, given a 1-hour rainfall total, in all three areas. This return period overestimation is dramatic in Italy where a higher 1-hour rainfall maxima total is expected. The Table 2 summarizes the return

time period estimates for 1-hour rainfall accumulation for the three study areas, derived from $R(Z_h)$, $R(Z_h, K_{dp})$ and from raingauges.

| Return period | Italy A1 | | | Italy A2 | | | Estonia | | |
|---|---|---|---|---|---|---|---|---|---|
| | WR-KDP | WR-ZH | RG | WR-KDP | WR-ZH | RG | WR-KDP | WR-ZH | RG |
| 2 | 41 | 23 | 34 | 37 | 20 | 29 | 21 | 17 | 17 |
| 10 | 70 | 41 | 56 | 58 | 35 | 47 | 34 | 30 | 28 |
| 20 | 81 | 47 | 64 | 66 | 40 | 54 | 39 | 35 | 33 |
| 50 | 95 | 56 | 75 | 77 | 48 | 64 | 45 | 41 | 38 |
| 100 | 105 | 63 | 83 | 85 | 53 | 71 | 49 | 46 | 42 |

**Table 2.** The return time periods estimates for 1-hour rainfall accumulation for the three study areas, derived from $R(Z_h)$, $R(Z_h, K_{dp})$ and from raingauges

The maxima 1-hour accumulation for a given return time obtained from $R(Z_h, K_{dp})$ shows better agreement with values obtained by raingauges. In the two Italian areas, the underestimation by $R(Z_h)$ is evident and larger in area A1 than in area A2. For Estonia, $R(Z_h, K_{dp})$ confirms good performance, while $R(Z_h)$ confirms underestimation but it also shows an unrealistic large-scale parameter and a low statistical significance of the fit. Recalling the findings in Voormansik et al. (2021a), QPEs derived by $Z_h$ show underestimation during the warm season. However, several reasons can explain the weakness of $R(Z_h)$:

1. horizontal radar reflectivity attenuation caused by intense instantaneous rainfall rates;

2. partial beam-filling;

3. inappropriate reflectivity cap threshold (55 dBZ) to avoid hail contamination. Disdrometer measurements reported cases of rainfall rates greater than 100 mm/h;

4. clutter residual;

5. inappropriate drop size distribution assumed to convert weather radar horizontal reflectivity in rain rate.

On the other side, $R(Z_h, K_{dp})$ estimations outperform in strongly convective precipitations regime like in Northern Italy, being immune from hail contamination and rainfall attenuation.

The major advantages of using weather radars for such applications are that information for unmeasured locations can be obtained and spatial gradients of the variables of interest captured. Due to the limited polarimetric weather radar data availabil-
ity in time (a few years), the present study is limited to climatological homogeneous areas, limiting or losing these advantages. Nevertheless, previous studies (Overeem et al., 2008, 2009a, b, 2010; Marra and Morin, 2015; Panziera et al., 2018; Marra et al., 2022) analysed weather radar QPEs based on horizontal reflectivity data adjusted with some ground raingauges measurements. Here, the major innovative aspect is that the QPEs, based on the blended algorithm $R(Z_h, K_{dp})$, are obtained independently

from co-located raingauges data availability. This study demonstrates that, by having polarimetric rainfall estimates, it is possi-
ble to estimate the rainfall annual maxima even in un-gauged regions. Moreover, as stated by Marra and Morin (2015), dealing
with QPEs based on radar reflectivity factor data, the upper threshold used to limit hail contamination is an issue in rainfall
maxima estimation in warm regions, limiting the instantaneous rainfall estimation typically to about 100 mm/h. Involving $K_{d}p$
for QPEs, the hail contamination issued is overcome, making QPEs independent from neither climatic region nor weather radar
attenuation. Future studies will benefit from longer time series allowing investigations in wider non-homogeneous areas.

## 5 Conclusions

Several studies investigated rainfall annual maxima derived from weather radarbased QPEs obtained by the traditional $Z_h - R$
relationship with some adjustments with raingauges. In the past decades, dual polarization weather data radar observations
are becoming available from operational weather radars. As stated by Bringi and Chandrasekhar (2001) and Voormansik et al.
(2021a), the benefits of using dual polarization variables like $K_{dp}$ in rainfall estimates are evident: these QPEs are immune
from weather radar miscalibration, anomalous propagation, and partial beam-blockings or beam-filling.
For the first time, this study investigates QPEs based on polarimetric observations by operational C-band weather radar located
in Italy and Estonia. The most remarkable aspects of this study are:

- data are derived from operational C-band weather radar without any dedicated settings;

- QPEs are derived by polarimetric observations without any raingauges adjustments.

As shown by Voormansik et al. (2021a), rainfall estimations based on $Z_h - K_{dp}$ algorithms are robust and reliable, overcom-
ing most of the sources' uncertainties: hence, no corrections nor adjustments with raingauges have been applied. The annual
maximum of 1-hour rainfall accumulation is typically assumed to have a general extreme value (GEV) distribution. Given the
shortness of weather radar data, this study is limited to a short duration (one hour) in homogeneous regions assuming parent
Gumbel distributions. Hence, Gumbel distribution parameters and depth-frequency curves have been derived from the 1-hour
dual-pol weather radar-based annual rainfall maxima. The comparison of weather radar return period estimations with ones
derived from gauge observations showed a good agreement. This study demonstrates that thanks to weather radar's high spatial
resolution, even a limited time series of weather radar observations can provide reliable estimations of extreme values distri-
bution parameters for annual hourly rainfall maxima in climatological homogeneous regions. It is worth recalling that QPEs
based on $R(Z_h - K_{dp})$ observations can be obtained only in cases of warmseason precipitations (anyway, when most intense
precipitations occur). The shown results demonstrate good agreement between QPEs obtained by $R(Z_h - K_{dp})$ and raingauges
data and consistent estimations of Gumbel distribution parameters. Assuming homogeneous regions with high-quality weather
radar observations, it is shown that even limited time-series weather radar observations can discriminate 1-hour rainfall total
annual maxima between different precipitation regimes. These results are promising especially if we recall that the two areas in
Italy are characterized by slightly different precipitation regimes and the applied statistical analysis can describe them properly.
The main requirements for weather radar observations applying this approach consist of proper weather radar calibration, radar

visibility, and a limited beam-broadening united to a limited beam height above the ground.

As longer rainfall time series based on dual-polarized meteorological data will become available, more investigations on different rainfall durations in wider and non-homogeneous areas will be possible, allowing estimations of spatial gradients and evaluations of different statistical distributions. Sub-hourly precipitation extremes can determine a wide range of impacts on infrastructure, economy, and even health causing urban flooding, triggering landslides, flash floods, and heavy soil erosion. Hence, future works will focus on sub-hourly rainfall accumulation intervals, estimating GEV parameter distributions and deriving other significant return periods.

*Code and data availability.* The code used to conduct all analyses, raingauges and weather radar data used in this study are available by contacting the authors.

*Author contributions.* R.C., T.V., D.M. and P.P. contributed to the design and implementation of the research, to the analysis of the results and to the writing of the manuscript. All authors read and approved the manuscript.

*Competing interests.* The authors declare no conflict of interest.

*Acknowledgements.* This research has been supported by the Estonian Research Council (grant no. PSG202). All figures included in the study were produced by the use of Free and Open Source Software (i.e. Quantum GIS Geographic Information System - Open Source Geospatial Foundation Project, http://qgis.osgeo.org, and the R Project for Statistical Computing, https://www.R-project.org/).

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
