# Peer review of "Estimation of extreme precipitations in Estonia and Italy using dual-pol weather radar QPEs"

_Atmospheric Measurement Techniques, 2022_

## Author Response (AR1)

RC1: 'Comment on amt-2022-220', Anonymous Referee #1

This paper presents the first application of extreme value statistics to precipitation estimates based on dual polarisation (dual-pol) weather radar. The topic is relevant to the community and to the readers of this journal, and the contribution is timely because dual-pol technology have been shown to improve radar precipitation estimates but extreme value statistics from such data were so far limited by the lack of sufficiently long records of dual-pol observations.

However, I highlight some aspects that need revisions before this study is considered for publication. These revisions may alter the overall results and conclusions of the study.

Major comments:

1. The study makes use of strong assumptions on the homogeneity of the extreme precipitation statistics field: all the three parameters of the GEV distribution used to represent the annual maxima are assumed to be uniform within the examined areas. This clearly helps with the problem of parameter estimation. However, while the authors carefully address the issue of independence in the used data, they completely neglect the homogeneity assumption. This homogeneity assumption is quite strong for today's research and practice standards and is not supported by specific tests/analyses. Spatial variations of precipitation statistics at scales smaller than the examined domains are known and reported in a number of previous studies based on both rain gauge and weather radar data (e.g., https://doi.org/10.1088/1748-9326/ab98b4; https://doi.org/10.5194/hess-26-1659-2022; https://doi.org/10.5194/hess-26-1439-2022). Neglecting the presence of spatial variations in the extreme precipitation statistics violates the identical distribution assumption of the extreme value theorem which underlies the adopted approach. Ultimately, this may lead to erroneous estimates of the parameters (samples from different distributions are mixed) and misleading evaluations of their uncertainty (the large number of data points will lead to small uncertainties, as shown in table 1). Alternative approaches in which only some of the parameters are homogeneous and the others are location-dependent (often referred to as regionalisation approaches) are available (e.g., https://doi.org/10.1080/0262666910949251) and are expected to provide results with comparable levels of uncertainty but higher accuracy. Studies based on weather radar or satellite data using such approaches are available in literature (e.g., see https://doi.org/10.1016/j.jhydrol.2019.04.081).

*Thank you for the valuable comment. The authors improved the paper by addressing the above-mentioned key points. Regarding Italy, careful gauges data quality leads to the removal of some gauges. Then, the application of some regional homogeneity tests (L-moment homogeneity test and the Anderson-Darling (ADAD) rank test) supported the assumption of regional homogeneity. For Estonia, the recent work by Olsson (2022) confirms homogeneity for 1-hour rainfall total maxima. The spatial independence of the samples is assured by the short duration considered in this study. Future studies dealing with longer durations will need to face also the spatial independence of the samples.*

*Finally, being critical estimations of the shape parameter on the basis of short time-series data (5-year weather data), it has been assumed equal to zero. Future works analyzing longer datasets could face this problem. Moreover, early warning systems are based on low return periods (typically 10-20 yr), and the shape parameter affects the tail*

*of the distributions, i.e. for longer return periods.*

2. A proper benchmarking of the results of this study with respect to results based on single-pol measurements is missing. Many studies about the analysis of extreme precipitation statistics from single-pol weather radars are not discussed in the introduction. As a consequence the key scientific questions underpinning this study are not well presented and leave the reader with questions like: why do we need to use dual-pol technology for these applications? what are the expected advantages over single-pol? what kind of uncertainties/errors were found in previous results based on single-pol? is dual-pol able to reduce such uncertainties/errors?

*The authors addressed this point by comparing the results with R(Zh) estimations. The description has been completely revised, trying to clarify the questions arisen by the reviewer. Given the assumptions, the study demonstrates that for convective precipitations during the warm season, R(Zh, Kdp) estimations are more reliable and they can be used for rainfall maxima estimations.*

3. Although some concepts about extreme value theory are presented in a reasonable way in lines 219-225, the manuscript misrepresents many key concepts. Section 2.1.1 in particular should be rewritten
4. The manuscript is not optimally organised. The results section contains large parts with methodological aspects and the discussion section contains main results rather than discussion.

*Thank you for the valuable comment. The manuscript has been revised with a better organization of the Sections.*

5. The main objectives and results should be better stated in the abstract and text. The last sentence of the abstract lists two main results reached by the study as (a) "weather radar observations can provide a reliable QPEs compared to rain gauges" and (b) "even relatively short time series can provide reliable estimation of the rainfall return periods in climatological homogeneous areas". However, point (a) is a result already addressed by previous literature (also listed in the introduction) and is not actually directly presented as a main result of this paper. Additionally, the study comes short at addressing point (b), because of an improper application of methods for extreme value analyses (see comment 1) and because it lacks a benchmarking of the results against single-polarisation data, neither from the radars used in this study nor from previous studies based on single polarisation technology (see comment 2). The natural question raised by the title and abstract of the paper "can dual-pol estimates improve our quantification of extreme precipitation probability with respect to single-pol estimates?" is not answered by the study.

*Benchmarking against single polarization QPEs has been reported, comparing results for all three areas for R(Zh, Kdp) and R(Zh) resect raingauges. The study demonstrates for 1-hour rainfall accumulation maxima that QPEs based on Zh-Kdp have a better agreement with rain gauges estimate of rainfall maxima.*

Minor comments:

- line 14-15: "Single C-band polarimetric…" this sentence is not clear
- line 19-27: it is weird to start the manuscript with text about climate change when the study assumes stationarity and does not provide any further result or discussion about changes

*The text has been re-phrased. Obviously, a climatological study is beyond this work. The authors wish to underlay that given more heavy precipitation in the next years due to global warming, there is an increased interest in rainfall total maxima.*

- lines 32 and line 41: the reference to Peleg et al 2016 is a bit misplaced as these authors only focused on one radar pixel. Other more relevant studies could help detailing the state of the art (see above some suggestions among many). Also note that the correct citation is Peleg et al. 2018

*The reference has been removed and other relevant studies added.*

- lines 37-48: this part is a bit confusing, please rephrase it

    *Done*

- Figures 1 and 2: I think these two figures could be merged. The quality of the figure should be improved, for instance using similar graphics and symbols

    *Done*

- The division of the Italian domain into two areas tries to address the homogeneity issue described above. It definitely help is some way (although it is in my opinion insufficient), but should be better motivated and explained. Currently it is mentioned in lines 93-95, but no details are presented

*Description and motivation for study areas in Italy have been improved. Some statistical analyses have been performed to support the regional homogeneity hypothesis.*

- line 197-225: this part would better fit the methods section

*Done*

- Figure 4: it is not fully clear what is the difference between the second and third panel of the first row and between the second and the third panels of the second row. The term model-simulated data only appears here and in line 229 of the text.

*The second QQ plot does not add additional information. It has been removed.*

- Figure 5: some quantitative information on the results presented in this figure should be added, also in relation to the biases of the radar-estimated values with respect to rain gauges and with respect to non-polarimetric radar estimates.

*Quantitative information has been provided and discussed by adding Table 2, which shows 1-hour rainfall maxima for several return periods.*

- lines 285-287: this conclusion is underpinned by crucial assumptions (see above). These assumptions should be clearly mentioned

*Conclusions have been re-phrased and improved on the basis of the additional analysis and with the aim to increase clarity.*

RC2: 'Comment on amt-2022-220', Anonymous Referee #2

The manuscript by Cremonini et al. presents an interesting application of radar dual polarization QPE to analyze extrema of precipitation to compute return times. Most of existing studies are based on raingauge rainfall measurements or reflectivity-based radar rainfall estimation. Kdp is instead used and expected advantage in estimating maxima is related to the better performance in QPE, especially at C-band and for intense precipitation, with respect to the Z-R based estimation. I recommend publication, after revision.

Major comments:

After reading abstract and the introductive parts of the manuscript, I was expecting some comparisons with Zh or Zh-Kdp rain algorithms, not in terms of QPE, but in term of impact on the GEV analysis in order to pointed out the need or the benefits of using a dual-pol radar approach. Such evidence is not made clear by the manuscript. Maybe a comparison with a single-pol approach could be helpful.

*Benchmarking against single polarization QPEs has been reported, comparing results for all three areas for R(Zh, Kdp) and R(Zh) resect raingauges. The study demonstrates for 1-hour rainfall accumulation maxima that QPEs based on Zh-Kdp have a better agreement with rain gauges estimate of rainfall maxima.*

Radar QPE is affected by the choice of parameters of radar rainfall algorithms. Even the performance of the R-Kdp estimator, which, theoretically is marginally affected by the DSD variability, can be influenced by such parameterization. If I am not wrong, authors have use relationships from literature. Also, the Kdp estimation method (different methods are used for the two study areas) can have an impact on results. Could the manuscript discuss this point ?

*The agreement between weather radar-based retrieval has been deeply investigated in Voormansik et al., 2021. The work evaluated algorithms' performances.*

Minor issues:

Line 32: Zh is more precisely the equivalent reflectivity factor at horizontal polarization. Please specify that authors prefer using a shorter language.

*Corrected with improvement in description between at line 32*

Line 60: "…rainfall intensity estimations based on R(Zh,Kdp)", why not R(Kdp?)

*R(Kdp) derived from C-band weather radar for low rain rates is insensitive and then, affected by large errors. The algorithm R(Zh, Kdp) allows for overcoming these large errors by relying in Zh for precipitation regimes where rain attenuation at C-band is negligible.*

Line 141: Is the 3-dB variation of bias resulting from self-consistency consistent with technical issues occurred ?

*Some miscalibration occurred during some devices failures. The Zh bias is generally related to miscalibration of the radar.*

Line 149: About R(Zh,Kdp): Is Zh corrected for attenuation ?

*No, attenuation correction is not needed as Zh attenuation at C-band is negligible for low rain rates (< 25 dBZ).*

Line 205: It is not clear to me what is "z" in the formula and "Z" mentioned in the caption of Fig- 3.

*The semi-variograms have been derived from R(Zh, Kdp) rainfall estimations. Corrected both text and caption.*

Figure 4. Please define units of axes.

*Added units for x-axis of leftmost plot (mm/h). The other plots show dimensionless quantities like sample's number and quantiles.*

Figure 5. please specify in the caption the meaning of dash lines.

*Added to the Figure's caption.*

---

## Author Response (AR2)

**Response to the reviewers**

The authors wish to thank the referees for their remarks. Thank to their valuable comments the paper has been improved. In the following, the point-by-point response to the reviews.

**Referee #1**

*1. I start with a technical comment. Did the authors check how well the Gumbel distribution represents the skewness of the annual maxima in the three areas? This can be a second big assumption that needs to be adequately supported. The L-moments could represent a good metric for it. Should the skewness not be compatible with \xi=0, there are options to use a-priori values different from 0.*

The authors addressed this point performing discordancy measure test, homogeneity test and goodness-of-fit based on L-moments as suggested by Hosking and Wallis (1997) for all datasets and for Italy and Estonia. The results allowed to identify suspicious raingauges, confirmed the acceptable regional homogeneity and, finally, confirmed the Gumbel distribution as one of the possible theoretical statistical distributions for the areas.
All these results have been described and discussed in the additional paragraph "2.1.1 Data quality, homogeneity, and goodness-of-fit".

*2. My main concern remains the issue with homogeneity. The authors responded to this major comment by: (a) for the case of Italy, testing the homogeneity of the region with two methods (AD rank test and L-moments homogeneity test); (b) for the case of Estonia referring to a previous study (Olsson & al 2022) in which the 10-year return levels in Northern Europe are examined. While I have some specific concerns (see the points i and ii below), I can see how similar arguments could hold for justifying such a homogeneity approximation in some situations.*
*(i) Olsson & al actually show variations in the 10-year return level within the study area in Estonia (see Fig. 2 and 3 in Olsson & al 2022).*
*(ii) Viglione & al 2007 is used to justify the use of AD and L-moments test for homogeneity. For the case of slightly-skewed distributions (such as the Gumbel distribution eventually used here) these authors recommend the L-moments test. This raises 2 questions: why not using only this method? and Why not using any test over Estonia?*

L-moments method (Hosking and Wallis, 1997) has been applied confirming acceptable homogeneity both in Italy and in Estonia. Regarding the comment on Olsson et al., 2022, it is worth noting that the study area considered in this paper is smaller than the one analyzed by Olsson et al.: eastern raingauges are outside from the studied area.

*One should note that the homogeneity discussed by Viglione & al is within regional frequency analyses, i.e. assuming that some of the parameters of a distribution are homogeneous within the region, while some others are allowed to change. As I mentioned in my first review, assuming all the parameters are in common is quite (too?) strong in the current hydrological practice. This is particularly true in a study based on weather radar. What is the point of using weather radar when eventually one only estimates one distribution? The big advantages of using weather radars for such applications is to (a) get information for ungauged locations and to (b) capture spatial gradients in quantities of interest. But if we assume identical distribution we kill both these advantages. Can't we just pool all the rain gauge data together for this?To my view, this remains a critical point in this study.*

The major advantages of using weather radars for such applications are that information for unmeasured locations can be obtained and spatial gradients of the variables of interest captured. Due to the limited polarimetric weather radar data availability in time (a few years), the present study is limited to climatological homogeneous areas, limiting or losing these advantages.
Nevertheless, previous studies (Overeem et al., 2008, 2009a, b, 2010; Marra and Morin , 2015; Panziera et al., 2018; Marra et al., 2022) analysed weather radar QPEs based on horizontal reflectivity data adjusted with some ground raingauges measurements. Here, the major innovative

aspect is that the QPEs, based on blended algorithm R(Zh, Kdp), are obtained independently from co-located raingauges data availability. This study demonstrates that, by having polarimetric rainfall estimates, it is possible to estimate the rainfall annual maxima even in un-gauged regions. Moreover, as stated by Marra and Morin (2015), dealing with QPEs based on horizontal reflectivity data, the upper threshold used to limit the effect of hail is an issue in rainfall maxima estimation in warm regions, limiting the instantaneous rainfall estimation typically to about 100 mm/h. Involving Kdp for QPEs, the hail contamination issued is overcome, making QPEs independent from neither climatic region nor weather radar attenuation. Future studies will benefit from longer time series allowing investigations in wider non-homogeneous areas.

*3. My comment 3 was not answered nor considered in the revision. Section 2.1.1 remains affected by erroneous and inaccurate statements about the extreme value theorem. It is beyond my role to correct point by point this section. The authors should refer to a textbook, to previous papers, or to wikipedia (https://en.wikipedia.org/wiki/Fisher–Tippett–Gnedenko_theorem). Now that the authors moved to using a Gumbel distribution, there is probably less need for details about the limiting types of GEV.*

The Section has been completely revised.

**Referee #2**

*Line 65. The R(Zh,Kdp) algorithm used in the study is a combination of a R(Zh) and a R(Kdp) algorithm. Such algorithms have been called "optimal", "composite", or "blended". I suggest to rephrase ".... that R(Zh,Kdp) blended algorithms provides good quality QPEs."*

Corrected

*Line 116-117: suggested "the annual hourly precipitation maxima concerns years from 1988 to 2020. "*

Corrected

*Line 118: replace "studly" with "study"*

Corrected

*Line 135: the text refers to a "shape" parameter that i explained later. Suggest to add something like "see section …"*

Corrected

*Line 155-156: "When using the default window length of 35, ...." Please specifiy also the size of bis of the 35-bin windows.*

Corrected

*Line 163-166: There is a discussion about Z-Kdp consistency in hail that in general does not exist. Although the discussion makes sense for Kdp calculation, it is not clear from the text wheter hail is considered in the computation of precipitation statistics or not.*

We tested with various window lengths and found length 8 to be the optimal compromise between spatial resolution and smoothness. After the window length change, we obtained realistic looking precipitation fields but the overestimation compared to gauge values increased. This is because $\Phi_{dp}$ gradients became steeper as a result of the smaller window length. To mitigate this issue we first decreased the high_z (high limit for reflectivity to remove hail contamination) value from 60 dBZ used in Voormansik et al. (2021a) to 50 dBZ which is the lowest recommended value by Giangrande et al. (2013). Because overestimation was still evident we also reduced the Zh -Kdp self-consistency coefficient. As stated by Kumjian et al. (2019) the R(Zh -Kdp ) consistency relationships probably do not exist in hail and it is therefore recommended to reduce the weight of the self-consistency constraint in the case of hail (Reimel and Kumjian, 2021). We tested with various values and found a coefficient value of 0.9 to produce optimal results.

*Line 169: Please indicate where the coefficients or rain algorithms are taken from*

The references have been included.

*Line 173: I would still recommend "data are" instead of "data is"*

Corrected

*Figure 3: Please fix the title "Italty"*

The title has been fixed.

*Line 349: "the following table" please replace with "Table 2". In the table is not clear the meaning "0.5ex> 0.5ex"*

Fixed

---

## Author Response (AR3)

**Response to reviewer #1**

The authors wish to thank the referee for his valuable comments that greatly improved the paper. In the last version, the abstract and conclusions underline the innovative use of non-adjusted QPEs derived for polarimetric weather radar data. The authors agree with the reviewer that the present work is the first step on a long road towards deeper analysis and regional investigations in non-homogeneous areas: further steps will be allowed by longer time series data that will become available by polarimetric weather radar in the next years.

Finally, as remarked by the editor, the reference list has been updated according to the Copernicus Publications requirements.